# Exploring Preventive Healthcare in a High-Risk Vulnerable Population

**DOI:** 10.3390/ijerph19084502

**Published:** 2022-04-08

**Authors:** Trisha L. Amboree, Jane R. Montealegre, Kayo Fujimoto, Osaro Mgbere, Charles Darkoh, Paige Padgett Wermuth

**Affiliations:** 1Department of Management, Policy, and Community Health, The University of Texas Health Science Center School of Public Health, Houston, TX 77030, USA; trish.amboree@bcm.edu (T.L.A.); paige.m.padgett@uth.tmc.edu (P.P.W.); 2Department of Epidemiology, Human Genetics and Environmental Sciences, The University of Texas Health Science Center School of Public Health, Houston, TX 77030, USA; 3Dan L Duncan Comprehensive Cancer Center, Baylor College of Medicine, Houston, TX 77030, USA; jrmontea@bcm.edu; 4Department of Pediatrics, Baylor College of Medicine, Houston, TX 77030, USA; 5Department of Health Promotion and Behavioral Sciences, The University of Texas Health Science Center School of Public Health, Houston, TX 77030, USA; kayo.fujimoto@uth.tmc.edu; 6Disease Prevention and Control Division, Houston Health Department, Houston, TX 77054, USA; osaro.mgbere@houstontx.gov; 7Institute of Community Health, University of Houston College of Pharmacy, Houston, TX 77204, USA; 8Microbiology and Infectious Diseases Program, University of Texas MD Anderson Cancer Center UTHealth Graduate School of Biomedical Sciences, Houston, TX 77030, USA

**Keywords:** preventive healthcare, sexual health, preventive medicine, public health

## Abstract

This study describes preventive care behaviors and explores opportunities to deliver preventive sexual healthcare to a high-risk vulnerable population. Data from the National HIV Behavioral Surveillance (NHBS) system high-risk heterosexuals (HET) cycle (2019) in Houston, Texas, was used to describe preventive care utilization and assess the relationship between healthcare utilization and sociodemographic characteristics. More than 47% reported having no usual source of healthcare, and 94.6% reported receiving no non-HIV STI testing in the past 12 months. Additionally, many sociodemographic factors were associated with healthcare utilization and having a usual source of healthcare. Future efforts should be targeted at increasing preventive healthcare utilization among high-risk vulnerable populations as well as implementing more preventive sexual healthcare services in the community health centers where these populations most frequently encounter healthcare.

## 1. Introduction

The term “vulnerable populations” has historically been used in research to refer to populations comprising a disadvantaged portion of the community that require specific consideration and protection [1]. However, this also includes a broader population that is of low socioeconomic status, those who are underinsured, and those who are a part of racial/ethnic minority groups [1,2]. Health risks, specifically disease acquisition and treatment, of these populations are exacerbated by lack of access to healthcare and preventive services, as adequate healthcare access is vitally important to prevent and treat illnesses [2,3]. These populations are also at increased risk for poor health outcomes [2,4]. The use of preventive healthcare services, such as screening, testing, and vaccination, is multidimensional and has been associated with different factors [5]. Racial/ethnic minority populations have historically reported not having usual sources of medical care, no health insurance or under-insurance, and low socioeconomic status [5]. The vulnerability of these higher-risk populations contributes to the disparate burden of morbidity and mortality that racial/ethnic minorities, specifically Black/African American and Hispanic/Latinx populations, tend to carry in regard to chronic illnesses and other poor health outcomes [5]. Furthermore, this leads to differing health-seeking behaviors among populations that are at increased risk of poor health outcomes [5].

One widely recommended and available preventive health service is routine testing for human immunodeficiency virus (HIV) and other sexually transmitted infections (STIs). Globally, the vast majority of HIV infections occur in low- and middle-income countries [6,7]. In the U.S., high-risk racial/ethnic minority populations tend to carry a higher disease burden in terms of HIV and other STIs [8]. Specifically, those who identify as Black/African American accounted for 42% of all new HIV diagnoses in the U.S. in 2019, and 29% were among those who identified as Hispanic/Latinx [9]. STIs as a group are also more prevalent in the U.S. among racial/ethnic minorities [10,11], with these disparities being the result of social factors such as poverty, lack of employment, and low education [12]. Preventive healthcare utilization, such as routine HIV and other STI testing, could greatly help in reducing the disease burden by decreasing transmission within these communities.

Another available and recommended preventive health service is vaccination against human papillomavirus (HPV). While HPV, the most common STI globally, is nearly ubiquitous among sexually active adults, Black and Hispanic women exhibit higher rates of HPV-associated cervical cancer compared with women of other races/ethnicities [13]. Vaccine coverage against preventable infections, such as HPV, may reduce the risk of persistent HPV infections within these high-risk populations. While the HPV vaccine is recommended routinely for young people aged 9–26 years, the catch-up vaccine is licensed for adults through the age of 45 years, for whom joint decision-making between providers and patients is recommended [14].

Efforts to increase preventive healthcare utilization among high-risk populations have included identifying key barriers to healthcare access and recognizing access to health services as a key social determinant of health in Healthy People 2030; however, much work needs to be completed to actively reduce health disparities [15]. To our knowledge, a representative study assessing preventive healthcare utilization has not been conducted in a mostly racial/ethnic minority population living in medically underserved areas of high socioeconomic deprivation. Thus, the objective of this study was to describe preventive healthcare behaviors among a high-risk vulnerable population as well as to explore opportunities to deliver preventive healthcare, specifically regarding HPV vaccination and HIV and other STI testing behaviors, to high-risk vulnerable populations.

## 2. Materials and Methods

This study utilizes data obtained from the Centers for Disease Control and Prevention (CDC) National HIV Behavioral Surveillance (NHBS) system in Houston, Texas. The NHBS collects data every year in populations at high risk for HIV infection—specifically, men who have sex with men (MSM), people who inject drugs (PWID), and heterosexually active adults at increased risk for HIV (HET)—and uses a standardized, interviewer-administered survey instrument to gather information on participant demographics, sexual behaviors, alcohol and drug use history, HIV and other STI testing and use of prevention services, and health conditions as well as site-specific questions of interest [16,17]. The HET population was utilized in this analysis and represents a highly vulnerable population, as defined by living in a census track with high levels of economic deprivation and disproportionately high rates of HIV infection, with the majority being from racial/ethnic minority groups.

Respondent Driven Sampling (RDS) methods were used to recruit participants in the high-risk heterosexual population. This hard-to-reach population is not usually captured by traditional sampling methods; thus, to obtain an adequate sample, RDS utilizes participants’ social networks [18]. Furthermore, RDS allows the sampling of a population that does not have an existing sampling frame [19]. RDS methods specific to NHBS have been described in detail elsewhere [20,21,22]. Briefly, this sampling utilizes a social network-based recruitment method that begins with initial recruits or “seeds” who are identified before the start of data collection. After the seeds complete the study activities, they are asked to recruit up to five other people they know or associate with. These recruited persons then complete the study activities and recruit others. This adaptive sampling technique is commonly used to sample hidden populations, such as high-risk heterosexual populations, and yields efficient estimates [19,20,21,22,23]. NHBS participants received incentives for their participation in study activities, including monetary compensation for time spent completing the survey and for providing specimens for HIV and STI testing [16].

### 2.1. Study Sample

The current study utilized data from the NHBS–HET cycle with data collection from July to December of 2019. The target sample was made up of individuals aged 18–60 years who lived in Houston or Harris County, identified as male or female, had vaginal or anal sex with someone of the opposite sex in the past 12 months, and were able to complete the NHBS interview in English or Spanish. NHBS also defines heterosexually active adults at increased risk for HIV as having low household incomes, defined as at or below 150% of the poverty guidelines adjusted for geographic differences in the cost of living. However, for the purposes of this paper, we did not exclude those who did not meet the low-income definition because all participants lived in areas with high socioeconomic deprivation and increased HIV prevalence. Additionally, those who reported non-prescription injection drug use in the past 12 months were excluded from the study. Furthermore, males who reported having other male sexual partners in the past 12 months were excluded, as both these populations are assessed in a different NHBS cycle of data collection [16].

The current study’s population was a subset of eligible NHBS study participants. NHBS participants were included in this study if they were aged 18–60 years at the time of their interview and completed the NHBS interview in 2019. Participants were excluded if they did not meet NHBS inclusion criteria or did not complete the interview. A total of 591 NHBS–HET participants met these criteria. The study protocol was reviewed and approved as “exempt” by the Committee for the Protection of Human Subjects at the University of Texas Health Science Center at Houston. 

### 2.2. Measures

The main outcome, healthcare utilization, was determined by assessing whether participants had a usual source of healthcare and their last healthcare encounter. The presence of a usual source of healthcare was assessed by asking participants “*Is there a place that you usually go when you are sick, or you need advice about your health? Do NOT include internet web sites*”. Furthermore, the time from the last healthcare encounter was assessed by asking the participants “*About how long has it been since you last saw a doctor, nurse, or other health care provider about your own health?*”. Having a usual source of healthcare was categorized as no usual source of healthcare, clinic or healthcare center, or doctor’s office or HMO. The time from the last healthcare encounter was categorized as within the past 12 months, 1–2 years ago, 2–5 years ago, or more than 5 years ago.

Testing behaviors assessed included receiving an HIV test in the past 12 months, HIV testing frequency, HIV testing location, and receiving an STI test other than HIV in the past 12 months. HIV testing in the past 12 months was assessed by asking participants “*Was your most recent HIV test in the past 12 months, that is, since [fill with interview month] of last year?*”. HIV testing frequency was assessed by asking participants “*When did you have your most recent HIV test? Please tell me the month and year*”. HIV testing frequency was categorized as never tested, tested 0–6 months ago, tested 7–12 months ago, tested 13–24 months ago, tested 25–60 months ago, or tested more than 60 months ago. HIV testing location was assessed by asking participants “*When you got tested [if month & year of last test are known, fill with response] where did you get tested?*”. HIV test location was categorized as HIV counseling and testing site, HIV/AIDs street outreach or mobile unit, drug treatment program, correctional facility, family planning or obstetrics clinic, community health center, private doctor’s office, emergency room, hospital inpatient, at home, or unspecified other. Other STI testing in the past 12 months was assessed by asking participants “*In the past 12 months, that is, since <interview month, [interview year-1]>, were you tested by a doctor or other health care provider for a sexually transmitted disease like gonorrhea, chlamydia, or syphilis? Do NOT include tests for HIV or hepatitis*”. Uptake of the HPV vaccine was assessed by the item *“Have you ever received a shot that protects against HPV, for example Gardasil?”* and was categorized as yes or no in the analyses.

For the analyses, HIV testing in the past 12 months was categorized as yes or no. Other STI testing in the past 12 months was categorized as yes or no. The participants’ sociodemographic variables included age in years (continuous and ranged from 18 years to 60 years), sex (dichotomized as female and male), and self-reported race/ethnicity (categorized as non-Hispanic White, non-Hispanic Black, Hispanic, or non-Hispanic other). The non-Hispanic other category refers to participants who reported being Asian, Alaskan Native, or Pacific Islander. Furthermore, sociodemographic variables included education (categorized as less than a high school diploma, high school diploma or equivalent, or at least some college education), health insurance type (categorized as no health insurance, private health insurance, public health insurance, or some other insurance coverage), and poverty level (categorized as above the poverty level or below the poverty level). Poverty was determined by assessing the participant’s self-reported annual household income and the number of dependents that relied on that income and comparing it with federal poverty guidelines. Additionally, homelessness (categorized as never homeless, currently homeless, or previously homeless but not currently homeless) and incarceration status (categorized as never incarcerated, incarcerated but not within the past 12 months, or incarcerated within the past 12 months) were included.

For regression analyses, healthcare utilization was assessed by collapsing the variable for the time from the last healthcare encounter into two categories: had a healthcare encounter within the past 12 months or did not have a healthcare encounter within the past 12 months. The usual source of healthcare was assessed by collapsing the variable into two categories: has a usual source of healthcare and does not have a usual source of healthcare.

### 2.3. Statistical Analysis

Data were cleaned, prepared, and formatted in SAS 9.4 (SAS Institute, Cary, NC, USA), exported as a comma-separated value file, and converted into an RDS object for analysis in RDS Analyst [24]. Questions about network size and characteristics of participants’ networks were used to create population weights to account for sources of bias inherent to RDS methodology and to calculate population estimates and sample variances [25]. RDS Analyst was utilized to generate population prevalence estimates along with 95% confidence intervals and standard errors. Furthermore, population cross-tabulations were conducted in RDS Analyst. Bivariable and multivariable regression analyses were conducted in SAS 9.4 using the modified Poisson regression approach with the log link function and robust variance estimation clustered on the recruitment chain (SAS Institute, Cary, NC, USA) [26]. Estimates from regression analyses were RDS-adjusted using Gile’s sequential sampling weights with an estimated population size of 53,690. The unweighted multivariable estimates were also included. The PROC GENMOD was used to generate unadjusted and adjusted prevalence ratios and 95% confidence intervals to assess the association between sociodemographic characteristics and indicators of healthcare access and utilization. Sociodemographic variables included age, sex, race/ethnicity, education, health insurance type, poverty, homelessness, and incarceration status. All statistical tests performed were two-tailed with a probability value of 0.05 used as the threshold for declaring statistical significance.

## 3. Results

Table 1 summarizes the sociodemographic characteristics of this study population. The 591 respondents examined had an average age of 38.7 years, 55% were female, 78.3% were non-Hispanic Black, 52.5% had a high school diploma or equivalent, 51.3% reported having no health insurance, 83.5% had a household income below the poverty level, 21.3% were currently homeless, and 51.1% had a history of incarceration.

Table 2 describes the primary care utilization of this study population. Of the 591 respondents, 47.7% reported having no usual source of healthcare; however, 68.6% had a healthcare encounter within the past 12 months. The primary sources of healthcare among those who reported having a usual source were family planning and obstetrics clinics and community healthcare centers (70.2%) (data not shown in tables). Furthermore, 58.3% reported not having received an HIV test within the past 12 months, and 20.9% reported never being tested for HIV. Of those who had received an HIV test, 39% received their most recent test from a community health center. Additionally, 94.6% reported not being tested for other STIs in the past 12 months. Lastly, only 8.3% reported having received at least one dose of the HPV vaccine. In those who were age-eligible, only 11.5% had received at least one dose of the HPV vaccine (data not shown in tables).

Table 3 shows the weighted unadjusted prevalence ratios and the weighted and unweighted adjusted prevalence ratios from the modified Poisson regression models of healthcare utilization and sociodemographic factors. There were 10 recruitment chains with a minimum cluster size of 1 and a maximum cluster size of 320. Age, educational attainment, poverty, and incarceration history were not statistically significant in the bivariable models. However, age, education, and poverty were retained in multivariable models due to a priori knowledge [5]. In the weighted multivariable model, being female (*p* < 0.0001), of Black race/ethnicity (*p* = 0.04), having at least some college education (*p* = 0.01), and having a private or public health insurance plan (*p* = 0.0003 and *p* < 0.0001, respectively) were all significantly associated with a higher prevalence of having had a healthcare encounter in the past 12 months. On the other hand, having a history of homelessness but not being currently homeless (*p* < 0.0001) was significantly associated with a lower prevalence of a healthcare encounter in the past 12 months. 

Table 4 shows the weighted unadjusted prevalence ratios and the weighted and unweighted adjusted prevalence ratios from modified Poisson regression models for having a usual source of healthcare and sociodemographic factors. There were 10 recruitment chains with a minimum cluster size of 1 and a maximum cluster size of 317. Although age, sex, race/ethnicity, educational attainment, poverty, and homelessness did not reach statistical significance in the bivariable models, age, sex, race/ethnicity, educational attainment, and poverty were retained in multivariable models due to a priori knowledge [5]. In the weighted multivariable models, reporting private or public health insurance coverage (*p* < 0.0001) was significantly associated with a higher prevalence of having a usual source of healthcare. By contrast, those with a history of homelessness but who were not currently homeless (*p* = 0.005), those with any incarceration history (*p* = 0.0003), and those who were incarcerated within the past 12 months (*p* = 0.005) were significantly associated with a lower prevalence of having a usual source of healthcare. 

## 4. Discussion

The findings from this study show a severe dearth of preventive healthcare utilization in this high-risk population. Almost half of the population reported having no usual source of healthcare, yet 68.6% reported having a healthcare encounter in the past year. We suspect that acute care facilities, such as emergency rooms, urgent care, and community health centers, may be utilized more often by these populations than doctor’s offices, which is supported by the finding that higher-risk populations tend to report having no usual source of healthcare and no consistent location for healthcare needs [5]. Furthermore, less than half of the population had received an HIV test in the past 12 months, and 95% had not received any other STI testing in the past 12 months. A study conducted by Kates et al. suggests that most HIV testing is performed in the private setting [27]; thus, the low utilization of testing in our study may also be due to not having a usual source of healthcare.

In the adjusted regression analyses, we found that Black participants had a higher prevalence of recent healthcare utilization compared with White participants. This finding was not expected but may be due to a lack of heterogeneity, as most of the study participants were Black (78.3%). Our findings also suggest a positive association between the prevalence of recent healthcare utilization and being female, having at least some college education, and having health insurance. This is consistent with the literature, as females tend to access care more often than men [28]. This may especially be observed with regard to preventive care [29]. The literature also supports our finding that those with health insurance coverage have higher healthcare utilization, as those who are uninsured tend to use fewer preventive services [30], and those who are less educated tend to report worse general health [31,32,33]. We did not find a statistically significant association between having a recent healthcare encounter and poverty. This finding may be due to a lack of heterogeneity with regard to poverty, as over 83% of our study population lived below the poverty level. 

The results showed that those who had any type of health insurance had a higher prevalence of having a usual source of healthcare. This is consistent with literature that suggests that those with health insurance tend to receive greater preventive healthcare coverage, more screening services, and more appropriate and timely utilization of these services [34]. Our finding that those who had been but were not currently homeless had a lower prevalence of having a usual source of healthcare is also consistent with literature that suggests that people with experiences of homelessness are often disengaged from primary services [35]. The finding that having any experience with incarceration resulted in a lower prevalence of having a usual source of healthcare is also consistent with the literature. The majority of our study population (68.3%) reported having some incarceration history; thus, there is an immense need to increase primary care access among those with a history of incarceration. The literature suggests that incarceration rates are higher among those of racial/ethnic minority groups and those with lower levels of education [36]. The high prevalence of incarceration history in our study population further elaborates the multilevel vulnerability of this population. Additional research is needed to effectively improve healthcare access for this population, specifically those with a history of incarceration. 

The largest proportion of those who had a usual source of care reported the source as either a clinic or a community health center. Thus, there may be an opportunity to introduce other preventive sexual health services at these centers. Data from the National Association of Community Health Centers suggest that, on average, community health centers serve one in three low-income uninsured persons [37]. Additionally, those who frequent community healthcare centers are disproportionately members of racial/ethnic minority groups, such as Hispanic and Black populations [37]. Therefore, healthcare encounters at community health clinics may be the only place where preventive healthcare can occur. Further research is needed to better understand how to increase preventive healthcare, specifically HIV and other STI testing and HPV vaccination, among higher-risk populations, along with the gap that community health centers can help fill.

### Limitations

This was a secondary analysis of a dataset that was not initially intended to examine healthcare utilization as a primary outcome. Thus, we may be missing other indicators of healthcare access and utilization, such as employment status, living in a rural community, lack of transportation, proximity to healthcare services, language barriers, and existing chronic conditions, as well as other contributing factors [38]. Additionally, there may be selection bias and other inherent biases because of the use of RDS methodology, which have been described elsewhere [25]. Our analyses are robust and account for the RDS sampling methodology; thus, we believe our results to be accurate estimates of the target population. The NHBS–HET cycle only captures data on male- or female-identifying persons; thus, other gender identities are not included, which may limit the generalizability of the results. In addition, the cross-sectional nature of the data limits the conclusions that can be drawn in relation to risk or causation [39]. Furthermore, the use of interview data increases the risk of information biases, such as recall bias and response bias. However, NHBS uses a CDC standardized questionnaire, which decreases this bias and increases the internal validity of this study [40]. The data should be interpreted with caution, as the attitudes that underlie the reported behaviors could not be ascertained. Notably, self-reported vaccination status has been shown to be racially biased; therefore, there may be some inherent bias with the use of self-reported vaccination as a marker or indicator of preventive healthcare utilization [41,42]. Lastly, the indicator used to estimate the HPV vaccine uptake in this study focuses on receiving one or more doses of the vaccine rather than vaccine completion.

## 5. Conclusions

In conclusion, preventive healthcare utilization, specifically HPV vaccine coverage and routine testing for HIV and other STIs, is extremely low in this high-risk, low-income population. Furthermore, almost half of the population reported not having a usual source of healthcare. Health insurance coverage, having experiences with homelessness, and having any history of incarceration were associated with whether a respondent reported having a usual source of healthcare, whereas sex, race/ethnicity, education, health insurance coverage, and having experiences with homelessness were associated with the time from the last healthcare encounter. Future efforts should be targeted at increasing preventive healthcare utilization among high-risk, low-income populations, specifically those with any history of incarceration and homelessness, as well as implementing more preventive sexual healthcare services in community health centers where these populations most frequently encounter healthcare.

## Figures and Tables

**Table 1 ijerph-19-04502-t001:** Demographic characteristics of the study population.

	N	Weighted % (95% CI)	SE
Age, Continuous (mean, SD)	591	---	38.7, 12.7
Sex			
Male	290	45.0 (37.5–52.4)	3.8
Female	301	55.0 (47.6–62.5)	3.8
Race/Ethnicity			
White	39	5.1 (2.2–8.0)	1.5
Black	465	78.3 (72.1–84.6)	3.2
Hispanic	75	14.9 (9.6–20.3)	2.7
Other	10	1.7 (0.1–3.3)	0.8
Education			
Less than HS diploma	155	27.1 (20.9–33.2)	3.1
HS diploma or equivalent	282	52.5 (45.7–59.3)	3.5
Some college or above	154	20.4 (15.6–25.2)	2.4
Health Insurance Type			
No health insurance	309	51.3 (44.7–58.0)	3.4
Private plan	39	6.8 (4.2–9.4)	1.3
Public plan	228	40.3 (33.8–46.7)	3.3
Other	9	1.6 (0.2–3.0)	0.7
Poverty			
Above poverty level	104	16.5 (12.1–20.9)	2.2
Below poverty level	487	83.5 (79.2–87.9)	2.2
Homelessness			
Never homeless	364	67.1 (60.1–74.0)	3.6
Currently homeless	135	21.3 (15.3–27.4)	3.1
Previously but not currently homeless	91	11.7 (8.1–15.2)	1.8
Incarcerated			
Never incarcerated	163	31.7 (25.7–37.6)	3.0
Incarcerated, but not within past 12 months	303	51.1 (44.4–58.0)	3.5
Incarcerated within past 12 months	125	17.2 (13.0–21.4)	2.1

Abbreviations: CI = Confidence Interval; SE = Standard Error; SD = Standard Deviation.

**Table 2 ijerph-19-04502-t002:** Testing behaviors and primary care utilization among the study population.

	N	Population % (95% CI)	SE
Usual Source of Healthcare			
No usual source of healthcare	291	47.7 (40.8–54.5)	3.5
Clinic or healthcare center	203	34.2 (28.1–40.3)	3.1
Doctor’s office or HMO	86	18.1 (13.6–22.7)	2.3
Last Healthcare Visit			
Within past year	394	68.6 (62.5–74.8)	3.1
1–2 years ago	103	14.7 (10.9–18.5)	2.0
2–5 years ago	76	14.5 (9.5–19.3)	2.5
5+ years ago	16	2.3 (0.7–3.8)	0.8
HIV Test in Past 12 Months			
Yes	223	41.7 (34.9–48.7)	3.5
No	358	58.3 (51.3–65.1)	3.5
HIV Testing Frequency			
0–6 months	125	27.5 (20.3–34.8)	3.7
7–12 months	88	17.3 (12.2–22.4)	2.6
13–24 months	86	18.7 (12.7–24.6)	3
25–60 months	55	8.2 (5.4–11.1)	1.5
60+ months	39	7.4 (4.4–10.4)	1.5
Never tested	114	20.9 (15.9–26.0)	2.6
Most Recent HIV Test Location			
HIV counseling and testing site	12	2.5 (0.7–4.2)	0.9
HIV/AIDS street outreach or mobile unit	82	16.9 (11.1–22.9)	3.0
Drug treatment program	7	0.6 (0.2–1.1)	0.2
Correctional facility	55	9.0 (5.3–12.7)	1.9
Family planning or obstetrics clinic	19	4.8 (2.1–7.4)	1.4
Community health center	149	39.0 (30.4–47.5)	4.3
Private doctor’s office	27	6.7 (3.4–10.0)	1.7
Emergency room	19	4.8 (1.5–8.1)	1.7
Hospital (inpatient)	37	6.8 (3.8–9.9)	1.6
At home	3	0.8 (-0.2–1.8)	0.5
Other	33	8.2 (3.5–12.9)	2.4
Gonorrhea, Chlamydia, or Syphilis Test in the Past 12 Months			
Yes	32	5.4 (2.4–8.5)	1.6
No	559	94.6 (91.5–97.6)	1.6
HPV Vaccine Uptake			
Yes	50	8.3 (5.2–11.3)	1.6
No	541	91.7 (88.7–94.8)	1.6

Abbreviations: CI = confidence interval; SE = standard error; STI = sexually transmitted infection; HPV = human papillomavirus.

**Table 3 ijerph-19-04502-t003:** Unadjusted and adjusted weighted and unweighted modified Poisson regression models assessing the relationship between healthcare utilization and sociodemographic characteristics in the study population.

	Healthcare Utilization in the Past 12 Months					
Characteristic	PR (95% CI)	*p*-Value	aPR (95% CI) ^a^	Adjusted *p*-Value	aPR (95% CI) ^b^	Adjusted *p*-Value
Age, Continuous	1.00 (1.00–1.01)	0.52	1.00 (1.00–1.01)	0.12	1.00 (1.00–1.01)	0.08
Sex						
Male (Ref)	1.00		1.00		1.00	
Female	1.24 (1.12–1.38)	<0.0001 ****	1.23 (1.13–1.34)	<0.0001 ****	1.11 (1.05–1.18)	0.0004 ***
Race/Ethnicity						
White (Ref)	1.00		1.00		1.00	
Black	1.13 (0.95–1.34)	0.17	1.13 (1.01–1.26)	0.04 *	0.97 (0.93–1.01)	0.13
Hispanic	1.09 (0.86–1.38)	0.49	1.10 (0.95–1.26)	0.20	0.95 (0.86–1.05)	0.32
Other	1.00 (0.47–2.15)	0.99	0.95 (0.44–2.05)	0.91	0.69 (0.41–1.15)	0.15
Education						
Less than HS diploma (Ref)	1.00		1.00		1.00	
HS diploma or equivalent	0.93 (0.74–1.17)	0.53	0.97 (0.83–1.14)	0.75	0.99 (0.93–1.06)	0.87
Some college or above	1.11 (0.90–1.38)	0.32	1.22 (1.04–1.43)	0.01 *	1.14 (1.07–1.21)	<0.0001 ****
Health Insurance Type						
No health insurance (Ref)	1.00		1.00		1.00	
Private/other plan	1.38 (1.21–1.58)	<0.0001 ****	1.31 (1.13–1.52)	0.0003 ***	1.29 (1.10–1.52)	0.002 **
Public plan	1.53 (1.41–1.66)	<0.0001 ****	1.51 (1.43–1.60)	<0.0001 ****	1.54 (1.45–1.64)	<0.0001 ****
Poverty						
Above poverty level (Ref)	1.00		1.00		1.00	
Below poverty level	1.02 (0.87–1.19)	0.82	1.13 (0.89–1.44)	0.33	0.99 (0.88–1.11)	0.90
Homelessness						
Never homeless (Ref)	1.00		1.00		1.00	
Currently homeless	0.85 (0.69–1.06)	0.16	0.87 (0.70–1.08)	0.21	0.97 (0.86–1.09)	0.54
Previously but not currently homeless	0.80 (0.73–0.88)	<0.0001 ****	0.76 (0.69–0.84)	<0.0001 ****	0.87 (0.79–0.96)	0.007 **
Incarcerated						
Never incarcerated (Ref)	1.00					
Incarcerated, but not within past 12 months	0.96 (0.88–1.05)	0.36	---	---	---	---
Incarcerated within past 12 months	0.91 (0.74–1.13)	0.39	---	---	---	---

^a^ Model adjusted with RDS-weights; ^b^ model not adjusted with RDS-weights; abbreviations: PR = prevalence ratio; CI = confidence interval; aPR = adjusted prevalence ratio; Ref = referent group. Significance levels: * = *p* < 0.05, ** = *p* < 0.01, *** = *p* < 0.001, **** = *p*< 0.0001.

**Table 4 ijerph-19-04502-t004:** Unadjusted and adjusted weighted and unweighted modified Poisson regression models assessing the relationship between usual source of healthcare and sociodemographic characteristics in the study population.

	Had a Usual Source of Healthcare					
Characteristic	PR (95% CI)	*p*-Value	aPR (95% CI) ^a^	Adjusted *p*-Value	aPR (95% CI) ^b^	Adjusted *p*-Value
Age, Continuous	1.00 (1.00–1.01)	0.54	1.00 (0.99–1.00)	0.85	1.00 (0.99–1.00)	0.27
Sex						
Male (Ref)	1.00		1.00		1.00	
Female	1.13 (0.92–1.38)	0.26	0.99 (0.90–1.08)	0.76	1.04 (0.89–1.21)	0.62
Race/Ethnicity						
White (Ref)	1.00		1.00		1.00	
Black	1.08 (0.94–1.25)	0.29	0.97 (0.91–1.04)	0.37	0.94 (0.87–1.01)	0.10
Hispanic	1.11 (0.79–1.56)	0.54	0.97 (0.80–1.18)	0.76	1.00 (0.89–1.13)	0.94
Other	1.48 (0.88–2.49)	0.14	1.78 (0.78–4.03)	0.17	1.09 (0.82–1.44)	0.56
Education						
Less than HS diploma (Ref)	1.00		1.00		1.00	
HS diploma or equivalent	1.05 (0.88–1.26)	0.60	1.04 (0.94–1.15)	0.48	1.03 (0.96–1.10)	0.37
Some college or above	1.11 (0.89–1.39)	0.35	1.04 (0.83–1.30)	0.77	1.04 (0.96–1.11)	0.33
Health Insurance Type						
No health insurance (Ref)	1.00		1.00		1.00	
Private/other plan	2.89 (2.67–3.14)	<0.0001 ****	2.67 (2.21–3.21)	<0.0001 ****	2.16 (1.83–2.55)	<0.0001 ****
Public plan	2.54 (2.18–2.96)	<0.0001 ****	2.59 (2.24–3.00)	<0.0001 ****	2.24 (1.93–2.60)	<0.0001 ****
Poverty						
Above poverty level (Ref)	1.00		1.00		1.00	
Below poverty level	0.89 (0.64–1.23)	0.47	1.02 (0.86–1.22)	0.79	0.90 (0.82–0.99)	0.04 *
Homelessness						
Never homeless (Ref)	1.00		1.00		1.00	
Currently homeless	0.64 (0.47–0.87)	0.004 **	0.90 (0.80–1.01)	0.08	0.94 (0.84–1.06)	0.32
Previously but not currently homeless	0.76 (0.53–1.10)	0.15	0.89 (0.81–0.96)	0.005 **	0.93 (0.79–1.09)	0.37
Incarcerated						
Never incarcerated (Ref)	1.00		1.00		1.00	
Incarcerated, but not within past 12 months	0.77 (0.60–0.98)	0.03 *	0.69 (0.57–0.85)	0.0003 ***	0.73 (0.57–0.93)	0.01 *
Incarcerated within past 12 months	0.75 (0.63–0.90)	0.002 **	0.71 (0.55–0.90)	0.005 **	0.70 (0.48–1.04)	0.08

^a^ Model adjusted with RDS-weights; ^b^ model not adjusted with RDS-weights; abbreviations: PR = prevalence ratio; CI = confidence interval; aPR = adjusted prevalence ratio; Ref = referent group. Significance levels: * = *p* < 0.05, ** = *p* < 0.01, *** = *p* < 0.001, **** = *p* < 0.0001.

## Data Availability

The data used in this study are owned and governed by the Houston Health Department; thus, the data are not publicly available.

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
