# Peer review of "Exploring Preventive Healthcare in a High-Risk Vulnerable Population"

_ijerph, 2022, doi:10.3390/ijerph19084502_

Round 1

Reviewer 1 Report

The results are interesting and have been critically analysed.

Personally , i think that the study can be considereted with a representative sample.

Author Response

This paper described the relationship between preventive health care behaviors and the susceptibility to STI in the high-risk vulnerable population. The introduction provides sufficient background and includes many relevant references. The research design is appropriate and the methods are adequately described. In addition, the results are interesting and clearly presented.

Point #1: However, the sampling method has a selection bias accordingly the sample is not representative. I think it would be appropriate to add another method of population selection and increase the sample size. Considering that in the literature there are other studies conducted with greater accuracy and scientific rigor, it would be advisable to redefine the outcomes in relation to the results obtained and to introduce in the questionnaire the knowledge and attitudes that underlie these behaviors.

Response: Thank you for this feedback. Due to the nature and limitations of the NHBS data used for this study, it would be impossible to add another method of population selection in order to increase the sample size, as the data are collected according to CDC’s criteria for the NHBS project. The HET population is only surveyed every three years and due to COVID protocol changes, the next HET cycle data collection is now scheduled for 2024. Further, the NHBS questionnaire is standardized for each cycle of data collection, and questionnaire changes/revisions are required to go through rounds of ethics approval. Thus, any questionnaire changes/revisions would not be added until the next cycle of data collection. To address these issues, we have included in the limitations section that there may be selection bias in our population due to RDS methods. We have also stated that the data should be interpreted with caution as we could not ascertain the attitudes that underlie these behaviors (lines 316-318; lines 327-328).

Point #2: It would be interesting to know which STIs were included in this study, the full list of STIs should be shown in an additional table.

Response: We agree with the reviewer about providing the full list of STIs included. The STIs included were gonorrhea, chlamydia, and syphilis. The manuscript has been revised with “bacterial” STI testing data in Table 2 and added a footnote that specifies included STIs as gonorrhea, chlamydia, and syphilis.

Reviewer 2 Report

This manuscript described the relationship between preventive health care behaviors and the susceptibility to STI in the high-risk vulnerable population.

Questions and Comments:

  1. There are some questions listed in the “measures” section, are those questions are part of all questions? the full list of questions (or the main category of the questions with the important questions as examples) may be shown as supplementary table.

  1. In table 2, for “HIV testing frequency” part, Never tested” should be putted at the bottom of that category.

  1. What kind of other STI has been included in this study, the full list of STI should be shown

Author Response

Point #1: There are some questions listed in the “measures” section, are those questions are part of all questions? the full list of questions (or the main category of the questions with the important questions as examples) may be shown as supplementary table.

Response: The questions listed in the measures section are the list of main questions that were relevant to this study. NHBS captures data for multiple categories and the full-length questionnaire is roughly 200 pages, so it may not be feasible to show the full questionnaire as a supplementary table. However, we have included a citation (#17) in the Materials and Methods section that provides a reference to the published full questionnaire (in line 92).

Point #2: In table 2, for “HIV testing frequency” part, “Never tested” should be putted at the bottom of that category.

Response: Thank you for this feedback. AS suggested, we have moved “Never tested” to the bottom of the “HIV testing frequency” category in Table 2.

Point #3: What kind of other STI has been included in this study, the full list of STI should be shown.

Response: We agree with the reviewer about providing the full list of STIs included. The STIs included were gonorrhea, chlamydia, and syphilis. The manuscript has been revised with “bacterial” STI testing data in Table 2 and added a footnote that specifies included STIs as gonorrhea, chlamydia, and syphilis.

Reviewer 3 Report

Thank you for the opportunity to review your manuscript. The report highlights some important considerations for access to preventive healthcare for high-risk vulnerable populations.  

I have some specific comments either because there are issues that must be addressed or for you to take into consideration.

line 160. If the goal was to get a representative sample, why was sex dichotomized to female/male? Were these the only options, or only female/male respondents?

lines 258-259. “The findings from this study show a severe dearth of preventive healthcare utilization in this high-risk population, especially in preventive healthcare.” What does it mean to have a dearth of preventive healthcare utilization in preventive healthcare?

lines 275- 276. “females tend to assess care more often than men.” Do you mean access?

lines 323-324 “therefore there may be some inherent bias with the use of self-reported vaccination as a marker of indicator of preventive healthcare utilization.” Do you mean ‘as a marker or indicator of’?

Author Response

Point #1: Thank you for the opportunity to review your manuscript. The report highlights some important considerations for access to preventive healthcare for high-risk vulnerable populations. I have some specific comments either because there are issues that must be addressed or for you to take into consideration. line 160. If the goal was to get a representative sample, why was sex dichotomized to female/male? Were these the only options, or only female/male respondents?

Response: Yes, we agree with the reviewer that the sample would be more representative with the inclusion of all gender identities, however, the NHBS HET cycle only collects data on those who are male- or female-identifying. We have included a statement in the limitations section in addressing this (lines 321-323).

Point 2: lines 258-259. “The findings from this study show a severe dearth of preventive healthcare utilization in this high-risk population, especially in preventive healthcare.” What does it mean to have a dearth of preventive healthcare utilization in preventive healthcare?

Response:  We have revised the manuscript correcting this grammatical error by deleting the phrase “especially in preventive healthcare” to improve clarity.

Point 3: lines 275- 276. “females tend to assess care more often than men.” Do you mean access?

Response: Thank you for noticing this typographical error. We have changed it to the correct word “access”, as suggested.

Point 4: lines 323-324 “therefore there may be some inherent bias with the use of self-reported vaccination as a marker of indicator of preventive healthcare utilization.” Do you mean ‘as a marker or indicator of’?

Response 4:  We have revised the manuscript by including the word “or”, as suggested.

Round 2

Reviewer 1 Report

The sampling method has a selection bias accordingly the sample is not representative.

In literature, there are other studies conducted with greater accuracy and scientific rigour.

It would be interesting to know which STIs were included in this study, the full list of STIs should be shown in an additional table.

Author Response

Comment 1: The sampling method has a selection bias accordingly the sample is not representative. In literature, there are other studies conducted with greater accuracy and scientific rigour.

Response: Thank you for your comment. This is a valid concern. This paper focused specifically on vulnerable hard-to-reach, underserved populations that are not typically captured by traditional sampling methods. These populations have heightened vulnerability to poor health outcomes and do not typically have a sampling frame or any other way to sample them. Thus, adaptive sampling strategies such as respondent driven sampling (RDS) methods have been employed in recent years and are considered valid for these hidden vulnerable populations. We have revised the methods section of the manuscript addressing this concern and additional references that describe validity of the RDS methodology in detail have also been include. We believe that despite the potential bias associated with RDS, information obtained from this hard-to reach, vulnerable, and under-studied population adds significant value to the body of literature in this subject area.

Comment 2: It would be interesting to know which STIs were included in this study, the full list of STIs should be shown in an additional table.

Response: Thank you for your comment. We have clarified in the methods section that this was a global question asked to all respondents during the survey. All STIs covered in the survey have been included. We have also added a variable name for STI testing in Table 2. The survey question specifically asked about bacterial STI tests such as gonorrhea, chlamydia, and syphilis.